# An Overview of Clinical Conditions and a Systematic Review of Personalized TMJ Replacement

**DOI:** 10.3390/jpm13030533

**Published:** 2023-03-16

**Authors:** Sergio Olate, Víctor Ravelo, Claudio Huentequeo, Marcelo Parra, Alejandro Unibazo

**Affiliations:** 1Division of Oral and Maxillofacial Surgery, Hospital A.G.P., Lautaro 4811230, Chile; 2Center for Morphological and Surgical Studies (CEMyQ), Universidad de La Frontera, Temuco 4811230, Chile; 3Division of Oral, Facial and Maxillofacial Surgery, Universidad de La Frontera, Temuco 4811230, Chile

**Keywords:** TMJ, TMJ prosthesis, TMJ replacement, orthognathic surgery

## Abstract

The temporomandibular joint (TMJ) is a complex structure in the cranio-maxillomandibular region. The pathological changes of the joint cause deficiencies at different levels, making its replacement necessary in some cases. The aim of this article is to analyze the current indications, treatment and criteria, and follow-up using a systematic review and case series. A systematic review was carried out, identifying the indications for the use of a customized TMJ prosthesis and evaluating criteria and validation in the international literature. After review and exclusion, 8 articles were included with a minimum follow-up of 12 months. The age of the subjects was between 18 and 47 years old. In 226 patients, 310 TMJ prostheses were installed, 168 bilaterally and 142 unilaterally. In most of the articles, a good condition in the follow-up was observed, with a reduction in pain and better conditions of mandibular movement and function. TMJ prosthesis and replacement is a protocolized, defined, stable, and predictable procedure. Indications and criteria must be evaluated by specialists and patients related to the pathology involved in TMJ deformity or degeneration. Randomized research with an accurate diagnosis and follow-up is necessary to obtain the best indication for this treatment.

## 1. Introduction

TMJ pathology is a complex condition. The use of three-dimensional images allows for an anatomical diagnosis and virtual planning in the case of surgical treatment [1]. It is possible to reproduce study models and manufacture customized surgical guides and functional implants [2]. These specific implants are made for each patient and can be used for complete substitution of the temporomandibular joint (TMJ), reconstruction of the maxillofacial skeleton, and orthognathic surgery [3].

The replacement of the TMJ may be performed with a stock system or with a computer design. In both cases, TMJ replacement has to replace the ramus and condylar component and the fossa component using titanium screws for fixing [4]. Customized titanium prostheses, designed by computer (CAD-CAM) from a three-dimensional model, have been shown to improve patient function [5], distribute the tension in the prosthesis [6], and give stable long-term results [7].

The TMJ connects the cranium with the inferior third of the face and is responsible for essential functions in the facial skeleton. For this reason, degenerative pathologies of the TMJ cause an adaptation in the orofacial function and in the sagittal position of the mandible, creating overload and changes in facial appearance [8,9]. The mandible interacts with the base of the cranium through the temporomandibular joint, and with the maxilla by dental occlusion, so the role of the TMJ shows compromises in all aspects of the mouth and face.

Facial morphology can change with TMJ pathology. This condition will show deficiencies in facial appearance and is related to facial surgery requirements [10]. Orthognathic surgery could be included in the surgical treatment [11].

The aim of this research is to perform a review of the use of customized TMJ replacement and to perform a systematic review of the use and results of this technology.

### 1.1. Indication for TMJ Replacement

The mandibular condyle is a key area in facial growth and development. Jing et al. [12] indicated that the regulation of condylar growth is related to chondrocytes located in the different layers of the head of the mandibular condyle. Likewise, Buschang and Gandini [13] indicated that the growth of the mandibular condyle is significantly related to the growth of the mandibular ramus, the mandibular body, and finally the position of the mandible.

During its development, the mandibular condyle may be affected by different conditions, such as an altered mechanical load, which leads to an increase in the maturation of chondrocytes, an increase in bone volume, and an increase in the early mineralization of the mandible; thus, altered loads may consistently affect the morphology of mandibular growth [14]. Recent publications by our group have shown that the mandibular condyle will adopt a different position depending on facial characteristics and also on malocclusion, determining a relationship between these two conditions [15].

Early treatment of TMJ pathology is more efficient in reducing biological requirements and economic costs [16,17], and it is the best choice for the early detection of TMJ pathology. In the case of TMJ replacement, the indication may vary with clinical and functional conditions. Meurechy and Mommaerts [18] carried out an extensive review and concluded that TMJ replacement is indicated in cases of (1) multiple previous operations in the TMJ, (2) failure of reconstruction with autogenous bone, (3) ankylosis of the TMJ, (4) neoplasia, and (5) severe functional alterations in the TMJ. Two further indications are associated with (6) disease of the connective tissue and autoimmune diseases, and (7) inflammatory, infectious, or reactive alterations that present disagreement. Mercuri [19] also included an eighth indication associated with loss of the posterior vertical dimension and deficiencies in occlusion, determining that TMJ replacement is also indicated in cases of facial deformities.

In an extensive review, Bach et al. [20] reported that out of 348 patients included, the most important indication for TMJ replacement was arthritis (39.7%), followed by ankylosis (27.6%) and degenerative joint disease (14.7%). Dimitroulis [21] included a classification of TMJ pathology and showed that Category 5, the last stage in condylar degeneration, is a catastrophic condition and terminal disease of the TMJ with a regular indication for TMJ reconstruction and replacement.

In this sense, TMJ disease could be related to volume reduction, such as in some cases of autoimmune disease; volume augmentation, such as in cases of hyperplasia or volume deformity; or absence, such as in cases of severe damage by trauma, multiples surgeries, or malformations. In all cases, the facial condition can be affected, and the TMJ replacement could also be linked to orthognathic surgery.

### 1.2. Customized or Stock Prostheses

Both systems are currently available in the market. From a financial point of view, the standard prosthesis is less costly in terms of manufacturing than the customized item and can be delivered more quickly. These are important arguments when making a clinical decision on which system to use, considering the urgency with which it is needed.

In terms of postoperative function, Kanatsios et al. [22] indicated that there are no significant differences between the systems in postsurgical pain or mandibular movement. Carter et al. [23] indicated that there are no significant differences between customized and standard prostheses in postoperative function, except in the early return to a normal diet.

One of the most important differences in favor of a customized prosthesis is linked with improvements in surgery. The customized prosthesis fits local anatomical conditions exactly. Movement analysis of the “new condyle” can be carried out at the design stage, and the best type of fossa for this function can be used. On the other hand, a detailed analysis can also be made to choose the best sites for fixing screws, planning osteotomies, and positions of the components of the prosthesis. It is also possible to create customized surgical guides. Along the same line, maxilla mandibular osteotomies, such as in orthognathic surgery, can be planned together with the TMJ prosthesis to obtain the best movement for the facial components.

### 1.3. Clinical Conditions Related to Use of TMJ Replacement

(a)Volume and Size Augmentation of the Mandibular Condyle

In the case of a hyperplastic condyle in an adult patient, condylar hyperplasia or an osteochondroma creating facial asymmetry are usual diagnoses (Figure 1).

In adults, low condylectomy could be the preferent treatment to resolve the facial asymmetry and the pathological condition. TMJ reconstruction using customized manufacture can be an option, as shown in this case. A surgical guide as a cutting guide for the osteotomy area is a good choice to take control of the bone movement in the osteotomy and the position of the TMJ prosthesis, the maxillary osteotomy, or the chin osteotomy. The use of a customized plate can be included in the protocol; however, the use of a surgical guide for bone cuts and a guide for plate position can help when customized plates are not present (Figure 2). The surgical guide for plate position can be designed with holes to assist in the surgery, using one or two of them to guide the position of the noncustomized plates. The final position of the maxillary or chin movements can be obtained with a protocol that includes the use of the combined virtual planning and customized TMJ with 3D printing technology or using stock plates with the complete customized TMJ replacement (Figure 3).

(b)Volume and Size Reduction of the Mandibular Condyle

The final stage of osteoarthrosis and internal derangement of the TMJ is the reduction of the condylar surface. Usually, the condylar head is short and close to the condylar neck because of the complete resorption of the condyle. A loss in vertical height of the ramus is common, and the airway can be severely compromised (Figure 4).

Orthognathic surgery and TMJ replacement can be simulated in virtual planning using the algorithm of software (Figure 5). In all cases, Digital Imaging and Communications in Medicine (DICOM) is segmented using Mimics software (Materialise, Leuven, Belgium), and the algorithm is used to create the maxilla, mandible, TMJ, and the requirements for the plan to design the treatments. A dental arch is included using the intraoral scanner (TRIOS, 3Shape, Copenhagen, Denmark). Obtained STL files are used in the protocol. Virtual planning is performed with the company 3DSurgery (3DSurgery, São Paulo, Brasil) and the company Enterprises (Artfix Implants, Pinhais, PR, Brazil).

Facial rotation in the clockwise direction is common. In this case, the reconstruction of the face using orthognathic surgery and TMJ replacement could be a good option. The design of the TMJ replacement is performed in agreement with the new position of the mandibular and maxillary components (Figure 6 and Figure 7).

(c)Malformations, Pathology, or Trauma of the Mandibular Condyle

The condition related to the malformation or severe destruction of the TMJ has to be treated using reconstructive techniques. Customized TMJ can help to restore the facial contouring and the TMJ function. as shown in Figure 8 and Figure 9.

## 2. Materials and Methods

A systematic review was performed according to the Cochrane Handbook *for* Systematic Reviews of *Interventions*. It was reported according to the Preferred Reporting Items for Systematic Reviews and Meta-Analyses (PRISMA) [24].

The aim of this research was to find the response to the following question: is the customized prosthesis for TMJ replacement related to stable clinical results in a follow-up over 12 months? The PICO strategy was used for this systematic review, including:P: The study population was the patients with treatment using TMR replacement;I: The intervention was the full replacement of the TMJ using a customized prosthesis;C: Variables included were diet, pain, and mouth opening;O: Stability of the TMJ replacement for more than 12 months.

A systematic search was performed in the PubMed, Lilacs, Cochrane, Trip, and Scopus databases from 2005 (after FDA approval of the use of customized TMJ) to January 2023. The following search strategy was used: (((((Temporomandibular joint replace) OR (Temporomandibular joint prosthesis)) OR (Artificial temporomandibular joint)) AND (Custom-made temporomandibular joint prosthesis)) OR (Customized temporomandibular joint prosthesis)) OR (Personalized temporomandibular joint prosthesis).

The sample was composed of patients with surgery for customized TMJ replacement in one or two joints with a description of pain, diet, and open mouth. Articles with a minimum follow-up of 12 months were included.

The titles and abstracts were selected independently by two investigators to verify their eligibility. In the case of discrepancy, consensus was reached by discussion or consultation with a third reviewer. The references that seemed to fulfill the inclusion criteria were reviewed in full text by the same reviewers. Publications in English, Spanish, and Portuguese were included. Studies with a sample size lower than 10 patients and articles with animals were excluded. Mendeley 2.80.1 software (Reference Management, Elsevier, London, England) to perform the final review of the articles.

Two reviewers extracted the data and evaluated the methodological quality of the studies by means of a predefined and standardized data form. A pilot test was used to ensure the homogeneity of the criteria between the reviewers. The reviewers were not blinded to the authors or journals. Data included:(a)Authors, country, and year of article;(b)Study design;(c)Sample: number, age, and sex;(d)Number and type of TMJ prosthesis;(e)Surgical specification for the operation; complementary surgery;(f)Software and company;(g)Follow-up of the sample.

A second data collection included:(a)Presence of pain;(b)Change in diet;(c)Mandibular function.

The methodological quality of the randomized clinical trials was evaluated with the Effective Public Health Practice Project (EPHPP) [25], including an evaluation of selection bias, study design, confounding, blinding, methods of data collection, and withdrawal and dropouts. Each methodological component was classified as strong, moderate, or weak based on the information provided by each study.

## 3. Results

Using a systematic search, 316 articles were identified. After duplicating and excluding articles, 107 papers were selected for review of titles and abstracts. Seventeen articles were included for full-text review (Figure 10). All the articles included customized TMJ replacement on one or both sides (Table 1).

After evaluation of the 17 articles, 8 were excluded for having a sample under 10 subjects [26,27,28,29,30,31,32,33], and 1 article [34] was excluded for having a follow-up lower than 12 months (Table 2).

Eight articles were included in the descriptive analysis (Table 3). Five studies were retrospective, and three studies were prospective. The minimum follow-up was 12 months, and the maximum follow-up was 49.7 months. The age of the subjects was between 18 and 47 years old. In 226 patients, 310 TMJ prostheses were installed, 168 bilaterally and 142 unilaterally. In four studies, condylectomy and coronoid resection were used [35,36,37,38]; in two studies, only condylectomy was performed [39,40]; in one article, the osteotomy was performed in the glenoid fossa and in the mandibular ramus [41]; and in one study, an analysis of the orthognathic surgery was performed at the same time [35].

In all the articles, a 3D image was used (for the TMJ design). Biomet TMJ was used in three articles [36,37,41]; the TMJ Concepts System was used in two articles [35,41]; and OMX TMJ Prosthesis [39], TMJ Yang’s System [40], and DARSN TM joint prosthesis were used in the other articles [42]. In one article, the manufacturer was not described.

In terms of follow-up, all the articles showed a better condition of patients after 12 months. Six articles evaluated the pain before and after surgery, and a significant reduction in pain in the first month after surgery was observed [35,36,37,38,39,40]. One article [42] evaluated the quality of life (QoL) in a sample of 20 subjects with a positive response.

The maximum open mouth was evaluated in six articles [35,36,37,39,40,42], and the average was an augmentation from 8 to 13 mm. The six articles included an evaluation of the diet in the preoperative and postoperative stages and concluded that a significant improvement in the type of diet was observed. Pinto et al. [35] showed a reduction in terms of lateral movement by 60% after 40 months of follow-up. Schlabe et al. [41] showed minimum data observing only the diagnosis and surgical treatment, and the measurement after surgery was based on mandibular movement.

The eight selected articles were evaluated according to different items (Figure 11 and Figure 12). In terms of selection bias, a low risk in 100% of the articles was observed. In terms of study design and blinding, 100% of the articles showed a high risk of bias. Regarding the confounders, four articles (59%) were strong, and four articles (50%) were weak. Of the articles, 75% showed data collection and withdrawal dropouts with a low risk of bias. The final rating of the risk of bias assessment for all selected articles was weak.

## 4. Discussion

Virtual planning and 3D printing technology could help to perform TMJ replacement and orthognathic surgery in the same operation [43]. This is beneficial for the patient because the plan prior to surgical intervention and the preparation of surgical guides will reduce the operating time [44].

Both standard and customized TMJ prostheses now allow significantly greater maximum mouth aperture and involve considerably less postoperative pain [4,45]. Due to their great adaptability and advantages in operation, specific implants are being increasingly chosen for TMJ replacement. They show biocompatibility, resistance to fatigue under mechanical load, and minimal wear on the joint surface, and their absolute precision improves clinical results [46].

Abramowicz et al. [47] investigated whether a stock TMJ prosthesis could be adapted to the models of patients treated with customized joint prostheses, finding that 23% of the standard prostheses could not be adapted. In those that could, significant changes had to be made to the cranial base or the mandibular ramus, removing at least 3 mm of bone. In subjects whose anatomy was close to normal, stock prostheses required more intraoperative work, which increased operating times and the financial cost for the whole process (with stock prosthesis even being cheaper than customized prosthesis). Orthognathic surgery is increasingly being incorporated in patients with TMJ replacement due to the excellent aesthetic and functional results that can be achieved in a single operation [48,49].

Morphological alterations of the TMJ are usually linked with changes in facial morphology, which are normally treated with orthognathic surgery. Wolford et al. [50] combined orthognathic surgery with unilateral TMJ replacement in six patients with hemifacial microsomia, performing follow-up at 6 years and 3 months. None of the subjects presented signs of pain, and there was a significant correlation between mandibular mobility and the subjects’ diet.

The sequence carried out by our group initially includes uni- or bilateral TMJ replacement, followed by Le Fort I osteotomy with maxillary repositioning, and finally genioplasty if necessary. This achieves a normal midline and stable, functional mandibular movement in cases that undergo postoperative functional rehabilitation. In this sense, the systematic review showed that mandibular movements, reduction in pain, and stability of the treatment can be expected in cases of TMJ replacement.

Humphries et al. [51] performed TMJ replacement with a follow-up of 2 years and 15 days. All the patients experienced a significant improvement in their maximum mouth aperture, with a range of 30 to 40 mm, and improved quality of life in both functional and aesthetic terms. This personalized approach offers multiple benefits to patients. For one, it ensures that the TMJ replacement is a perfect fit for the patient’s anatomy, minimizing the risks of surgical complications. Additionally, since the TMJ replacement is created from the patient’s individual characteristics, it also ensures that the patient’s postsurgery recovery is much faster. The main disadvantage can be related to the economic cost and the timing necessary to design and build the customized system. Sembronio et al. [42] used a protocol including customized plates, increasing the cost of the treatment; perhaps the use of stock plates in orthognathic surgery could present a lower economic cost. On the other hand, the time involved in the operating room will be increased in the case of stock implants; therefore, the customized TMJ replacement can cover the initial higher economic cost using lower requirements in the OR and the hospital.

The articles included in this review showed no significant complications in the follow-up and the absence of pain. Analyses performed by Gerbino et al. [36] showed improvements in dental occlusion, mandibular movement, diet consistency, aesthetic appearance, and quality of life, so the use of TMJ replacement related to severe TMJ pathology could be included in the armamentarium of surgeons.

Long-term follow-up is necessary, especially in patients who undergo unilateral TMJ replacement combined with orthognathic surgery, since the nonreplaced TMJ may suffer dysfunction [52]. In a retrospective study of 70 subjects, Pérez et al. [53] evaluated the functioning of the contralateral joint of patients with unilateral TMJ replacement. They observed that in subjects with unilateral replacement, the nonreplaced TMJ showed a 30% chance to replace them with TMJ prosthesis in the future; on the other hand, 70% of the patients with no TMJ pathology continued free of pathology over time. A detailed diagnosis of both TMJs must therefore be carried out to ensure that the joint that is not operated on will remain in good health in the future.

There are some questions to resolve in the future. In the case of patients with TMJ dysfunction, the requirements for dental rehabilitation or bruxism could be performed by some types of devices [54], and this condition needs further analysis in patients with TMJ replacement. Along the same lines, in the case of disc displacement, as can occur in the nonreplaced TMJ, the use of an occlusal splint can be realized [55]; however, the effect in the prosthesis has not been studied, and this can be a challenging perspective on the full treatment of bilateral TMJ pathology.

Systemic conditions involved in TMJ dysfunction, such as autoimmune pathology [56], need to be included in the follow-up of patients submitted to unilateral or bilateral TMJ replacement. The terminal stage of the TMJ can be treated by customized TMJ prosthesis; however, in the case of unilateral replacement, the nonreplaced TMJ needs support to remain healthy. Tissue engineering, such as the use of stem cells, can help in disk or bone treatment using potential cell differentiation to help in chondrogenic/osteogenic differentiation or in modulations of the immune or inflammatory process [57].

Acri et al. [58] conducted an extensive review of tissue engineering for the temporomandibular joint, and they included some options for each anatomical area of the TMJ, evaluating the disease and the potential treatment in each case. They concluded the potential for the future and addressed the next challenge for research in the field. According to the authors of the present research, the use of customized implants in the TMJ and customized treatment with stem cells obtained from the patient can help in the integration and modulation to accelerate the process for better function in a short time.

Artificial intelligence (AI) is a novel technology that can be used for TMJ diagnosis and treatment. AI is used in software for the diagnosis, planning, and design of customized TMJ prostheses. Indeed, the algorithm for segmentation is included in this protocol.

Recently, AI has been included in using natural language to define the diagnosis of TMJ in nonexpert dentists, working with key symptoms assessed by experts and showing positive results in the training model [59]. In the same line, in another study, the convolutional neural network using cephalometric images has been included. The model was trained with 500 cephalometrics to find the ratio between facial deformity and degenerative TMJ disease. The authors concluded the high congruence between them, so their use in regular diagnosis can be included. The requirements and type of treatment may be the next challenge in the evolution of the model to obtain an objective indication for the use of nonsurgical or surgical treatment, and to define when the catastrophic disease of the TMJ will be treated rationally by prosthetic replacement [60,61].

In the authors’ opinion, integrative treatment using AI for diagnosis, 3D printing technology, and the use of regenerative medicine are the greatest challenges for TMJ replacement.

## 5. Conclusions

TMJ replacement is a protocolized, defined, stable, and predictable procedure. Indications and criteria must be evaluated by specialists and patients related to the pathology involved in TMJ deformity or degeneration. Randomized research with an accurate diagnosis and follow-up is necessary to obtain the best indication for this treatment.

## Figures and Tables

**Figure 1 jpm-13-00533-f001:**
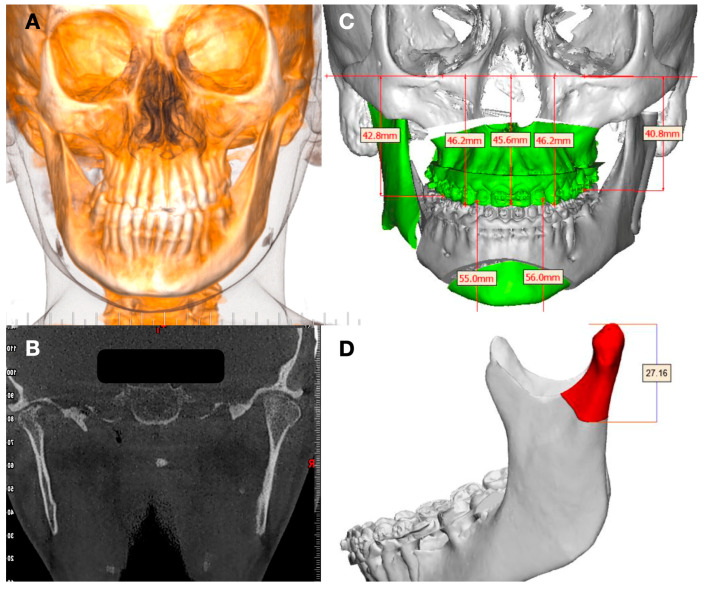
(**A**) Facial asymmetry related to overgrowth of left condyle. (**B**) CBCT showing large size of left condyle. (**C**,**D**) Facial plan to treat TMJ disease with facial asymmetry using low condylectomy.

**Figure 2 jpm-13-00533-f002:**
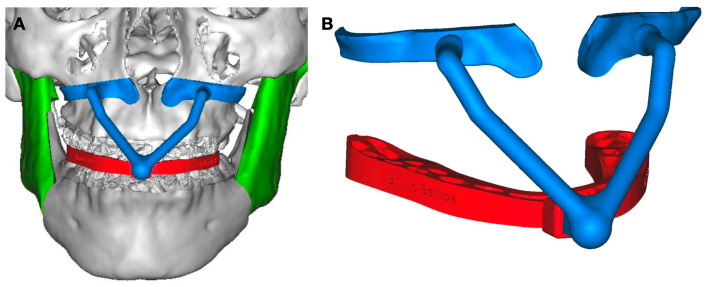
(**A**) Surgical guide for maxillary osteotomy was created and planned for dental occlusion after mandibular surgery. (**B**) Surgical guide for osteotomy taken as reference to perform osteotomy and is used for asymmetry control (design by 3DSurgery, São Paulo, SP, Brazil).

**Figure 3 jpm-13-00533-f003:**
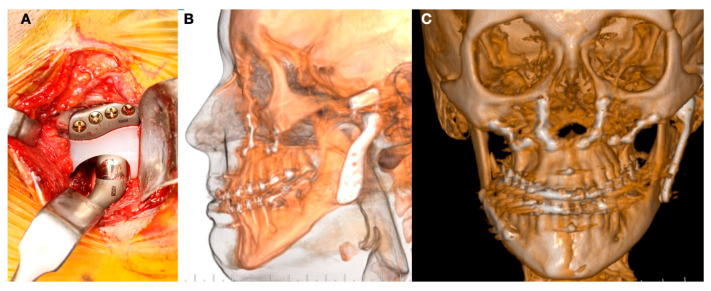
(**A**) Surgery to replace mandibular fossa and mandibular condyle (Enterprises, Artfix Implants, Pinhais, PR, Brazil). (**B**,**C**) Lateral and frontal view showing facial balance and symmetry after surgery.

**Figure 4 jpm-13-00533-f004:**
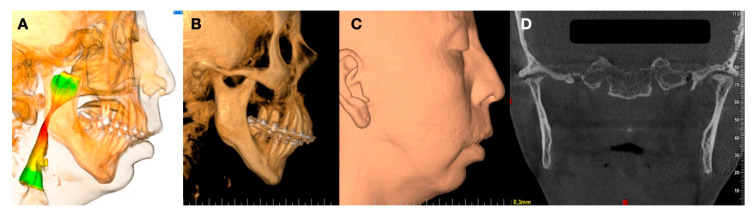
(**A**) Class II facial deformity with reduction in airway. (**B**) Maxillomandibular clockwise rotation with loss in vertical position of mandibular ramus. (**C**) Lack of facial balance and low sagittal projection of lower third (**D**) condylar disease related to stage 5 in TMJ degeneration with complete absence of condylar head.

**Figure 5 jpm-13-00533-f005:**
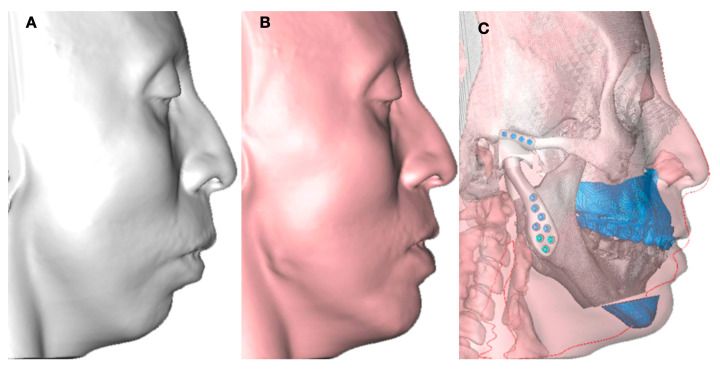
(**A**) Patients submitted to treatment using bilateral TMJ customized replacement and orthognathic surgery. (**B**) Superimposition of facial soft tissue using artificial intelligence of software and algorithm included in computational program. (**C**) Comparison of facial movement.

**Figure 6 jpm-13-00533-f006:**
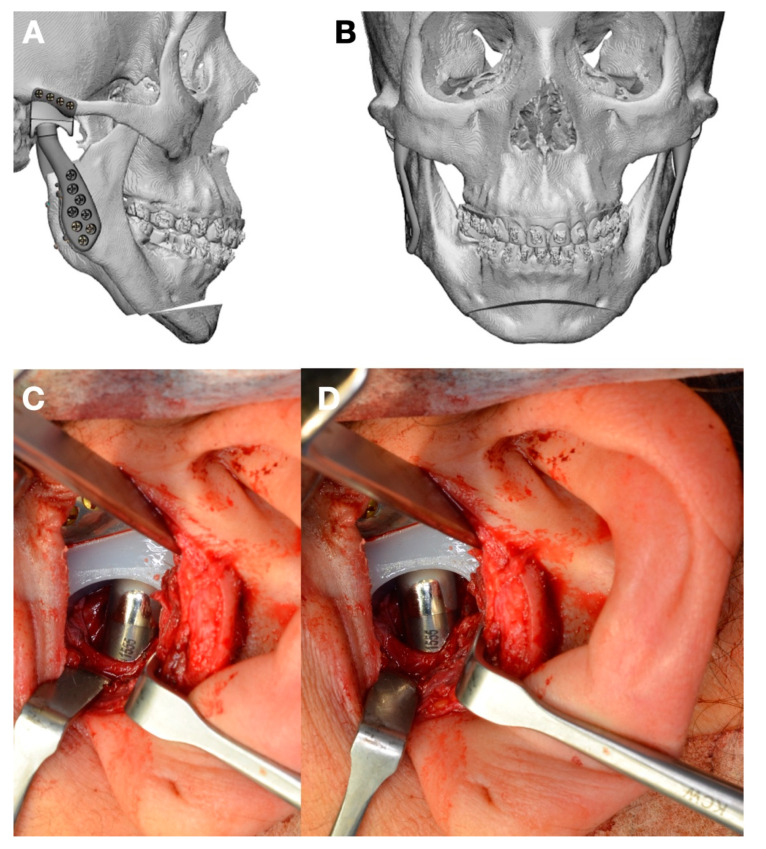
(**A**,**B**) TMJ design to obtain best position of mandibular condyle and ramus. (**C**) Condylar position of prosthesis in closed mouth. (**D**) Condylar position in open mouth (Enterprises, Artfix Implants, Pinhais, PR, Brazil).

**Figure 7 jpm-13-00533-f007:**
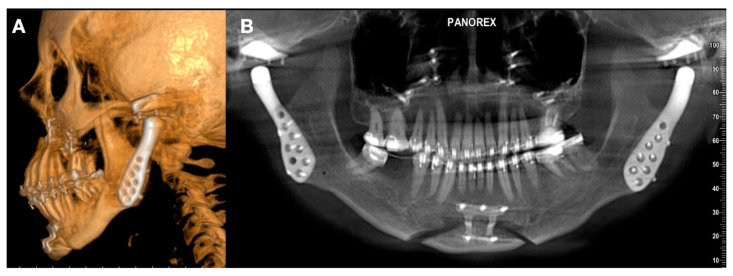
(**A**,**B**) Postoperative condition of TMJ replacement showing new position of occlusal plane related to counterclockwise rotation using orthognathic surgery and TMJ replacement.

**Figure 8 jpm-13-00533-f008:**
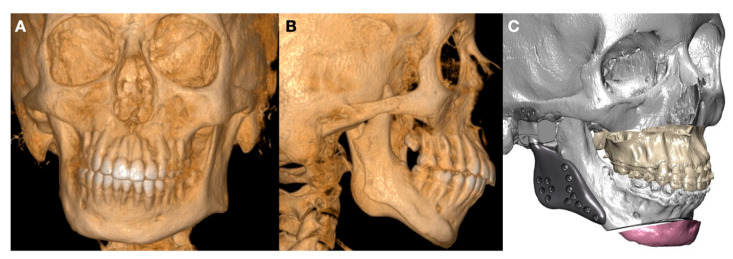
(**A**,**B**) Hemifacial microsomia type II of prusansky. Lack of TMJ anatomy, lack of facial balance and facial asymmetry, and deficiency in maxilla mandibular function can be expected. (**C**) TMJ reconstruction using TMJ replacement could be a good option in terms of function and aesthetic results, as well as stability over time (Enterprises, Artfix Implants, Pinhais, PR, Brazil).

**Figure 9 jpm-13-00533-f009:**
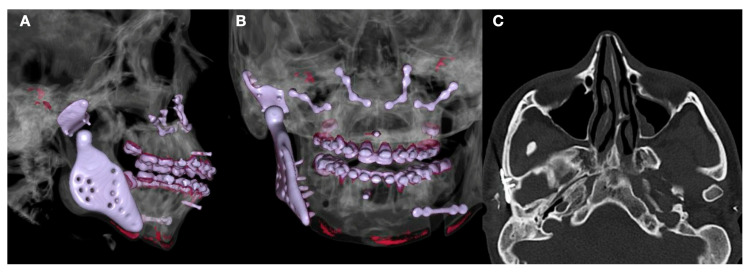
(**A**,**B**) TMJ replacement using customized TMJ prosthesis. (**C**) Screw in zygomatic arch and relations obtained with bone in this position.

**Figure 10 jpm-13-00533-f010:**
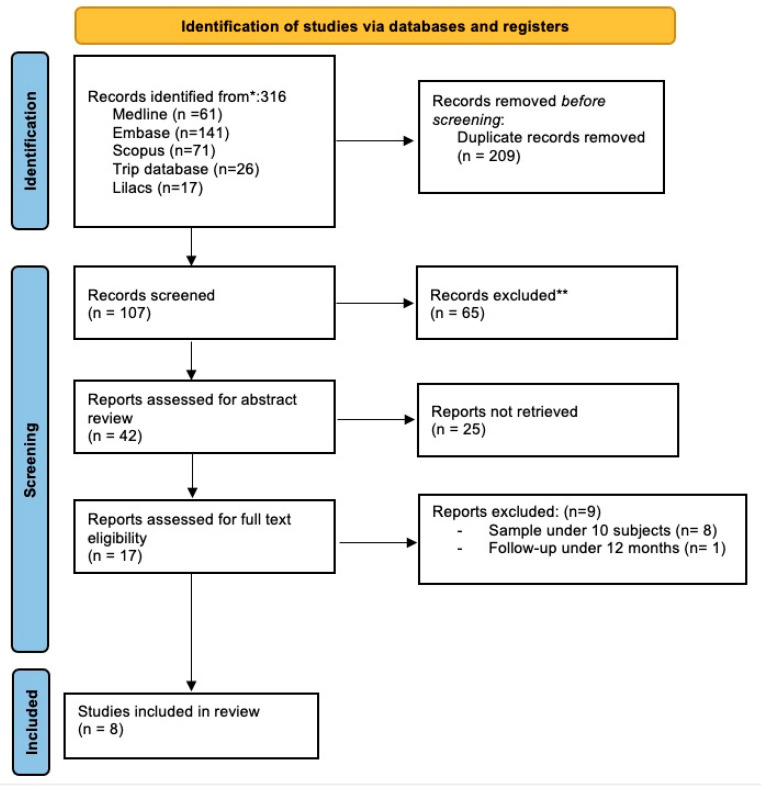
Flow chart of systematic review. * Records founded in the databases; ** Excluded duplicate articles by software.

**Figure 11 jpm-13-00533-f011:**
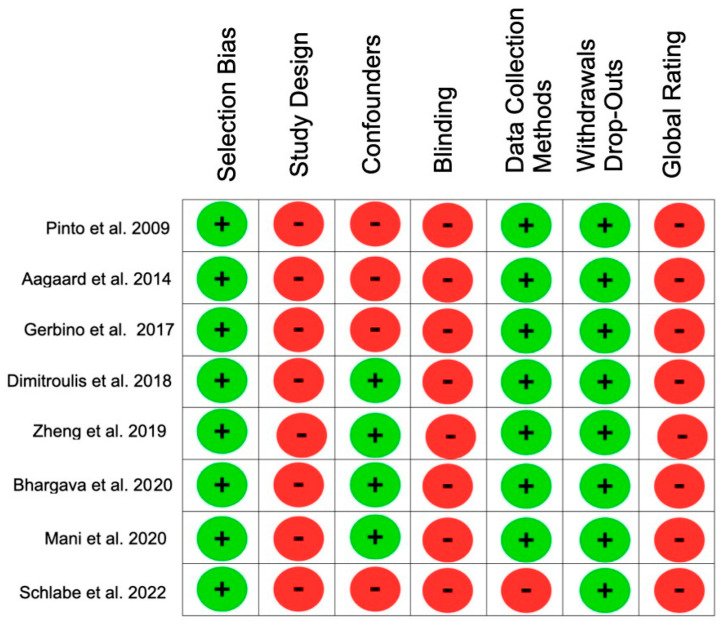
Summary of risk of bias of included studies (green: strong; yellow: moderate; red: weak).

**Figure 12 jpm-13-00533-f012:**
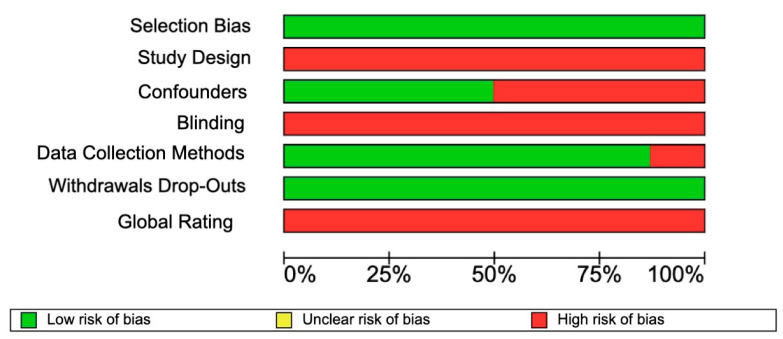
Summary plot risk of bias of 8 included studies.

**Table 1 jpm-13-00533-t001:** Characteristics of 17 articles in initial assessment used in systematic process to confirm indications (diagnosis), follow-up, and sample characteristics.

Author	Aim	N	Sex (M/F)	Age (Years)	Follow-Up (Months)
Pinto et al., 2009	To evaluate the pain and function after TMJ replacement and orthognathic surgery	47	47/0	34.5	40
Westermark et al., 2011	Study of the clinical conditions in a case series using TMJ replacement in extreme mandibular defects	4	4/0	32.7	72
Haq et al., 2014	To show a technique for ankylosis resection and reconstruction with customized TMJ prostheses	5	2/3	44.6	19
Aagaard et al., 2014	To describe the clinical variables after TMJ replacement	64	17/47	41.16	14.2
Gerbino et al., 2016	Retrospective study about TMJ replacement in TMJ ankylosis	6	6/6	44.3	18
Ciocca et al., 2016	A pilot study to use the TMJ replacement with fibula-free flap in oncologic disease	2	1/1	ND	60
Gonzalez-Perez et al., 2016	To evaluate the surgery, complications, pain, and mandibular movement in subjects treated by stock or customized TMJ replacement	7	19–38	51.8 ± 52.6	60
Gerbino et al. 2017	Retrospective study comparing stock vs. customized TMJ replacement	12	9/29	45.1	12
Dimitroulis et al., 2018	To show clinical data about the subjects treated with TMJ customized replacement in terminal TMJ pathology	38	7/31	43.8	15.3
Siegmund et al., 2019	To evaluate the clinical conditions after TMJ replacement in terms of mandibular movements, pain, and diet.	16	9/19	45.0	6
Zheng et al., 2019	To show the safety and efficacy of the design of customized TMJ prosthesis using 3D printing	12	5/7	47.8	12
Zheng et al., 2019	To confirm the evidence about the safety and efficacy of the TMJ replacement	5	2/3	42.4	13.8
Bhargava et al., 2020	To evaluate the efficacy of the customized TMJ replacement	20	12/8	28.75	12
Mani et al., 2020	The mandibular movement, clinical conditions in patients under growth, and follow-up related to growth, diet, and stability were evaluated	21	16/5	17.8	48
Chen et al., 2021	To show the results of the manufacturing of the TMJ under a standardized flow	9	0	56.56	12
De Sousa Gil et al., 2022	The aim was to show the TMJ reconstruction for complex TMJ disease using the intraoral and extraoral approaches	5	ND	ND	22
Schlabe et al., 2022	To show experience and the observations related to TMJ replacement	12	6/6	40.6	49.7

**Table 2 jpm-13-00533-t002:** Characteristics of 9 articles excluded. All articles used customized TMJ prosthesis and were excluded due to a sample size of less than 10 patients.

Author	N	Surgical Technique	Complementary Surgery	Side of Replacement	Manufacture	Planning SOFTWARE	Follow-Up (Months)
Westermark et al., 2011	4	Unilateral BSSO	LFI osteotomy in 2 patients	4 unilateral	ND	ND	72
Haq et al., 2014	6	Fossa osteotomy and condylectomy	ND	4 bilateral 1 unilateral	Biomet implants	ND	19
Gerbino et al., 2016	6	Condylectomy and osteotomy of the coronoid process	LFI osteotomy	6 bilateral	Biomet Microfixation	ND	18
Ciocca et al., 2016	2	Unilateral BSSO	ND	2 unilateral	ND	ND	60
Gonzalez-Perez et al., 2016	7	ND	ND	4 bilateral 3 unilateral	Biomet Microfixation	ND	60
Siegmund et al., 2019	16	Condylectomy	ND	16 unilateral	Zimmer Biomet	ND	6
Zheng et al., 2019	5	Fossa osteotomy, condylectomy, and osteotomy of the coronoid process	ND	5 unilateral	Arcan Al, MÖLNDAL	3-Matic Materialise	13.8
Chen et al., 2021	9	Condylectomy	ND	9 unilateral	ND	Mimics Materialise	12
De Sousa Gil et al., 2022	5	Unilateral BSSO	LFI Osteotomy in patient with bilateral replacement	3 unilateral 2 bilateral	Engimplan Medical Device	ND	22

**Table 3 jpm-13-00533-t003:** Characteristics of the 8 articles included for methodological assessment.

Author	N	Sex (M/F)	Age (Years)	Study Design	Surgical Technique	Complementary Surgery	Side of Replacement	Manufacture	Software	Follow-Up (Months)
Pinto et al., 2009	47	47/0	34.5	Retrospective	Condylectomy and coronoid process osteotomy	Orthognathic surgery	43 bilateral 4 unilateral	TMJ Concepts System	ND	40
Aagaard et al., 2014	64	ND	41.16	Prospective	Condylectomy and coronoid process osteotomy	ND	47 unilateral 17 bilateral	Biomet TMJ	Medical Modeling Inc.	14.2
Gerbino et al. 2017	12	9/29	45.1	Retrospective	Condylectomy and coronoid process osteotomy	BSSO	9 bilateral 3 unilateral	Biomet Microfixation TMJ	ND	12
Dimitroulis et al., 2018	38	7/31	43.8	Retrospective	Condylectomy	ND	12 bilateral 26 unilateral	OMX TMJ prosthesis	ND	15.3
Zheng et al., 2019	12	5/7	47.8	Prospective	Condylectomy	ND	12 unilateral	TMJ Yang’s system	3-Matic Materialise	12
Bhargava et al., 2020	20	12/8	28.75	Prospective	Condylectomy and fossa osteotomy	ND	18 unilateral 2 bilateral	DARSN TM joint Prosthesis	ND	12
Mani et al., 2020	21	16/5	17.8	Retrospective	Condylectomy and coronoid process osteotomy	ND	21 unilateral	ND	ND	48
Schlabe et al., 2022	12	6/6	40.6	Retrospective	Ramus and fossa osteotomy	ND	11 unilateral 1 bilateral	TMJ Concepts System of Zimmer Biomet	ND	49.7

## Data Availability

The data are available upon request from the corresponding author.

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
