# Peer review of "An Overview of Clinical Conditions and a Systematic Review of Personalized TMJ Replacement"

_jpm, 2023, doi:10.3390/jpm13030533_

Round 1

Reviewer 1 Report

dear colleagues, thank you for submission. After review and 18 exclusion, 8 articles were included. I agree the idea of personalized Replacement of Temporomandibular Joint (TMJ) has become increasingly viable. TMJ replacement is an important medical procedure that can help improve the quality of life of those suffering from TMJ related issues.  

Some comments to consider: 

1) add supp material for all search done 

2) line 57 add pico statement for problem and add hypotheses

3) unfourtunately meta analysis was not (could not performed) thus used SWIM https://www.bmj.com/content/368/bmj.l6890 in addition to PRISMA

4) provide working definition for 3d printing = Using 3D printing technology, TMJ replacements can be fabricated to a high level of accuracy and personalization. In discussion expand that computer-aided design tools allow for precision and speed in the designing and manufacturing process. Additionally, artificial intelligence algorithms can be used to detect the most suitable replacement for each individual patient, based on the patient’s unique characteristics. 

5) clinical implications are not well discussed - This personalized approach offers multiple benefits to patients. For one, it ensures that the TMJ replacement is a perfect fit for the patient’s anatomy, minimizing the risks of surgical complications. Additionally, since the TMJ replacement is created from the patient’s individual characteristics, it also ensures that the patient’s post-surgery recovery is much faster. I suggest to enlarge figures and annotate them. 

Follow the standard of Figures by  Semberiono - https://www.sciencedirect.com/science/article/abs/pii/S0901502720301338 (This is not my paper, but their visuals very neat)

In discussion I would like to see paragraph about how authors can suggest that personalized TMJ replacement could further be enhanced by the use of functional materials, such as cellular tissue engineered implants or electroactive materials. This would allow for the TMJ replacement not only to look like the patient’s original joint, but also to move and feel like it.  

https://www.mdpi.com/2073-4360/13/23/4175/htm

https://www.ncbi.nlm.nih.gov/pmc/articles/PMC7075314/

Thank you again for this important paper. 

Minor please replace , with . in numbers presentation 

I was happy to see the traffic light plot please add summary plot 

Author Response

1- Dear colleagues, thank you for submission. After review and 18 exclusion, 8 articles were included. I agree the idea of personalized Replacement of Temporomandibular Joint (TMJ) has become increasingly viable. TMJ replacement is an important medical procedure that can help improve the quality of life of those suffering from TMJ related issues.  

Dear reviewer, thank you for your help; the paper is better after your suggestions. All of them were included in the new manuscript.

2- line 57 add pico statement for problem and add hypotheses

PICO statement was included

3- unfourtunately meta analysis was not (could not performed) thus used SWIM https://www.bmj.com/content/368/bmj.l6890 in addition to PRISMA

The meta was no realized because the results of the review.

4-  provide working definition for 3d printing = Using 3D printing technology, TMJ replacements can be fabricated to a high level of accuracy and personalization.

the change was realized.

6- In discussion expand that computer-aided design tools allow for precision and speed in the designing and manufacturing process. Additionally, artificial intelligence algorithms can be used to detect the most suitable replacement for each individual patient, based on the patient’s unique characteristics. 

discussion was expanded to included topic as CAD CAM, AI and regenerative therapy

7- clinical implications are not well discussed - This personalized approach offers multiple benefits to patients. For one, it ensures that the TMJ replacement is a perfect fit for the patient’s anatomy, minimizing the risks of surgical complications. Additionally, since the TMJ replacement is created from the patient’s individual characteristics, it also ensures that the patient’s post-surgery recovery is much faster. I suggest to enlarge figures and annotate them. 

the statement was included and a discussion was included over them

Follow the standard of Figures by  Semberiono - https://www.sciencedirect.com/science/article/abs/pii/S0901502720301338 (This is not my paper, but their visuals very neat)

some figures were modified to get better resolution (300dpi) and the protocol as suggested by Semberiono was included in the images as well as in the discussion to talk about the economical cost involved in the process.

8- In discussion I would like to see paragraph about how authors can suggest that personalized TMJ replacement could further be enhanced by the use of functional materials, such as cellular tissue engineered implants or electroactive materials. This would allow for the TMJ replacement not only to look like the patient’s original joint, but also to move and feel like it.  -https://www.mdpi.com/2073-4360/13/23/4175/htmhttps://www.ncbi.nlm.nih.gov/pmc/articles/PMC7075314/

papers were added, and discussion was performed.

9- Thank you again for this important paper. 

Thank you sir.

10- Minor please replace , with . in numbers presentation 

change was realized

11- I was happy to see the traffic light plot please add summary plot

the figure was included

Reviewer 2 Report

Dear Authors your article is interesting, but it is necessary to review some points 

Materhials you must to add the PICO questions of your review.

In the results section you must to describe the risk of bias of your studies adding the % for single bias Adding the section “Quality assessment and risk of bias” i suggest you some lines to write this section.
Using the RoB 2, the risk of bias among the RCTs analyzed was estimated and reported in Figure …. Regarding the randomization process, ….% of the studies ensured a low risk of bias. Only …..% of RCTs excluded a performance bias, but …..% reported all outcome data, and half of included trials adequately left out bias in selection of the reported results. Overall, only … out of …. RCTs demonstrated a low risk of incurring bias.

- The introduction section is very short and is needed to add other references to increase the quality of the manuscript, Preferably a published articles should be with 90 or more references.

Suggested (https://doi.org/ 10.3390/prosthesis4020025) ; (doi: 10.1097/SCS.0000000000008771) (doi: 10.1097/SCS.0000000000009103)
(DOI: 10.1080/08869634.2022.2137129
)

You need to review the grammar of your article.

I suggest you add a table with the list of abbreviations used in the text.

-Please expand conclusion section with main results and future perspectives of this study

Figures are blurry. Please provide a higher-resolution file.

Thank You,

Kind Regards

Author Response

1- Dear Authors your article is interesting, but it is necessary to review some points. Materhials you must to add the PICO questions of your review.

Dear reviewer: thank you for your suggestion and help in this article. PICO statement was included.

2- In the results section you must to describe the risk of bias of your studies adding the % for single bias Adding the section “Quality assessment and risk of bias” i suggest you some lines to write this section.
Using the RoB 2, the risk of bias among the RCTs analyzed was estimated and reported in Figure …. Regarding the randomization process, ….% of the studies ensured a low risk of bias. Only …..% of RCTs excluded a performance bias, but …..% reported all outcome data, and half of included trials adequately left out bias in selection of the reported results. Overall, only … out of …. RCTs demonstrated a low risk of incurring bias.

change was realized.

3- The introduction section is very short and is needed to add other references to increase the quality of the manuscript, Preferably a published articles should be with 90 or more references. Suggested (https://doi.org/ 10.3390/prosthesis4020025) ; (doi: 10.1097/SCS.0000000000008771) (doi: 10.1097/SCS.0000000000009103)
(DOI: 10.1080/08869634.2022.2137129)

articles were included, and discussion was modified.

4- You need to review the grammar of your article.

the article was revised by medical paper translator company; if you consider, we can submit to another company.

5- I suggest you add a table with the list of abbreviations used in the text.

the list was included

6- Please expand conclusion section with main results and future perspectives of this study

Discussion was modified, and perspectives were included.

7- Figures are blurry. Please provide a higher-resolution file.

Figures was modified and now they have 300dpi